# Liver Transplantation for T2 Hepatocellular Carcinoma during the COVID-19 Pandemic: A Novel Model Balancing Individual Benefit against Healthcare Resources

**DOI:** 10.3390/cancers13061416

**Published:** 2021-03-19

**Authors:** Umberto Cillo, Alessandro Vitale, Michael L. Volk, Anna Chiara Frigo, Paolo Feltracco, Annamaria Cattelan, Giuseppina Brancaccio, Giuseppe Feltrin, Paolo Angeli, Patrizia Burra, Sara Lonardi, Silvia Trapani, Massimo Cardillo

**Affiliations:** 1Hepatobiliary Surgery and Liver Transplantation Unit, Department of Surgical, Oncological, and Gastroenterological Sciences, Padua University Hospital, 35128 Padua, Italy; cillo@unipd.it; 2Division of Gastroenterology and Hepatology, Loma Linda University Health, Loma Linda, CA 92354, USA; MVolk@llu.edu; 3Biostatistics, Epidemiology and Public Health Unit, Department of Cardiac, Thoracic, Vascular Sciences and Public Health, University of Padua, 35128 Padua, Italy; annachiara.frigo@unipd.it; 4Section of Anesthesiology and Intensive Care, Department of Medicine—DIMED, Padua University Hospital, 35128 Padua, Italy; paolo.feltracco@aopd.veneto.it; 5Infectious Diseases Unit, Department of Internal Medicine, Padua University Hospital, 35128 Padua, Italy; annamaria.cattelan@aopd.veneto.it (A.C.); ggbrancaccio@gmail.com (G.B.); 6Regional Transplant Center, 35128 Padua, Italy; trapiantifegato@aopd.veneto.it; 7Unit of Internal Medicine and Hepatology, Padua University Hospital, 35128 Padua, Italy; pangeli@unipd.it; 8Multivisceral Transplant Unit, Gastroenterology, Department of Surgery, Oncology and Gastroenterology, Padua University Hospital, 35128 Padua, Italy; burra@unipd.it; 9Veneto Oncologic Institute, IRCCS, 35128 Padua, Italy; sara.lonardi@iov.veneto.it; 10Italian National Transplant Center, Italian National Institute of Health, 00118 Rome, Italy; silvia.trapani@iss.it (S.T.); massimo.cardillo@iss.it (M.C.)

**Keywords:** liver transplantation, T2 HCC, COVID-19 pandemic, organ allocation, transplant benefit, waiting list harm

## Abstract

**Simple Summary:**

Using Italian national data from more than 8000 patients, we proposed a novel model calculating the net benefit of liver transplantation (individual benefit minus harm to others on the waiting list) in T2 hepatocellular carcinoma (HCC) patients in different scenarios of transplant activity reduction. Our results show that the transplant net benefit is closely related to the decrease in the number of organs, but it is also higher in T2 HCC patients than in non-HCC patients with the same model for end-stage liver disease (MELD) scores. Our model supports liver transplantation for T2 HCC with the highest net benefit also during crises such as COVID-19.

**Abstract:**

The COVID-19 pandemic caused temporary drops in the supply of organs for transplantation, leading to renewed debate about whether T2 hepatocellular carcinoma (HCC) patients should receive priority during these times. The aim of this study was to provide a quantitative model to aid decision-making in liver transplantation for T2 HCC. We proposed a novel ethical framework where the individual transplant benefit for a T2 HCC patient should outweigh the harm to others on the waiting list, determining a “net benefit”, to define appropriate organ allocation. This ethical framework was then translated into a quantitative Markov model including Italian averages for waiting list characteristics, donor resources, mortality, and transplant rates obtained from a national prospective database (n = 8567 patients). The net benefit of transplantation in a T2 HCC patient in a usual situation varied from 0 life months with a model for end-stage liver disease (MELD) score of 15, to 34 life months with a MELD score of 40, while it progressively decreased with acute organ shortage during a pandemic (i.e., with a 50% decrease in organs, the net benefit varied from 0 life months with MELD 30, to 12 life months with MELD 40). Our study supports the continuation of transplantation for T2 HCC patients during crises such as COVID-19; however, the focus needs to be on those T2 HCC patients with the highest net survival benefit.

## 1. Introduction

Although liver transplantation (LT) is theoretically the best therapy for patients with hepatocellular carcinoma (HCC), its main limitation remains the great discrepancy between the transplant demand and the supply of donor organs [1]. This dramatic imbalance has imposed the choice of clear allocation principles for patients with and without HCC, such as utility, urgency, or transplant benefit. HCC patients are mainly selected according to criteria focused on avoiding a poor post-LT outcome (i.e., utility), while non-HCC cirrhotic patients are usually prioritized for LT based on their risk to die without transplant, which is measured by the model for end-stage liver disease (MELD) score according to the so-called “sickest first” policy (i.e., urgency) [2]. In the last 20 years, the adoption of different principles for HCC and non-HCC patients and the attempt to balance these two populations using arbitrary scores has created relevant inequalities in resource distribution [3]. Transplant survival benefit is defined as life expectancy with transplantation minus life expectancy without transplantation. It is a very interesting principle with the potential to offer a fair distribution of resources between HCC and non-HCC patients [4,5,6]. In some countries, applications of transplant survival benefit are already present in current clinical practice to define, for example, a minimum threshold value of transplant eligibility (i.e., MELD 15) [5], or to decide eligibility, prioritization, and allocation [7,8]. However, none of the available criteria for individual clinical decisions for LT take into consideration population characteristics in terms of healthcare resources and waiting list (WL) features.

The need to link individual clinical decisions to population/healthcare system characteristics has been stressed by the current COVID-19 pandemic. The COVID-19 pandemic has dramatically hit many health systems worldwide, inducing a substantial reallocation of structural, personnel, and financial resources to the infection frontline [9,10,11,12,13]. Due to their complex and multi-interfaced functions, the COVID-19 pandemic has particularly impacted solid organ transplant (SOT) programs [14,15,16]. In countries with high rates of COVID-19 infection, such as Italy, Spain, and the US, organ donation has substantially decreased. According to early data from the Italian national center for transplantation, a 25% to 40% reduction of donor availability and transplant activity was observed from the end of February throughout March 2020 in Italy [17]. Similar figures have been reported in Spain, with a reduction in transplants of up to 40%, and in the US, with a reduction up to 80% [18,19]. The diversion of many professionals dedicated to donation processes to COVID-19-related needs, ICU patient overload, and severe hospital logistic distress represents the most relevant determinant behind such dramatic activity reductions [20,21]. Although, in many countries, COVID-19 vaccination campaigns are increasing, faster-spreading variants of the SARS-CoV-2 coronavirus show that recurrent outbreaks of SARS-CoV-2 are highly probable in the near future [22].

Common sense solutions have been diffusely put in place [8]. However, there is a substantial lack of solid scientific analysis to guide clinical decisions, especially in LT, where a huge variety of indications and clinical presentations have to be managed. In particular, whether it is appropriate to transplant T2 HCC patients in these situations of acute organ shortage remains controversial. T2 HCC (single nodule 2–5 cm, or 2–3 nodules ≤ 3 cm) now represents the vast majority of listed HCC patients [23], reaching almost 40% of overall transplant indications in Italy [6,24]. These patients, however, do not have an immediate risk of dying without LT, thus, in situations of acute organ shortage, many countries have limited their resources only to the most urgent patients [8,14]. A four-dimensional model of quadripartite equipoise that models ethical tensions in transplant activity in response to the increasing burden on healthcare systems has been recently proposed [25]. This model, however, generates a quite arbitrary semiquantitative score that has the potential to represent a guide for the regulation of transplant activity in a general population (i.e., fluctuation of the score over time guides the need to pursue or limit transplant activity), but it is not designed for individual clinical decision-making. 

The aim of this study was to provide a quantitative and flexible model to aid this decision-making in LT for T2 HCC in different scenarios of transplant activity reduction. The model quantifies individual transplant benefit and the expected harm to others on the WL due to individual organ allocation, considering different possible scenarios of transplant activity reduction. This model also has the potential, therefore, to be used to calculate the net benefit of LT (individual LT benefit minus harm to others on the WL) in T2 HCC patients.

## 2. Materials and Methods

### 2.1. Philosophical/Ethical Framework

We designed a conceptual ethical framework intended to address the ethical tension arising from the individual need for LT in T2 HCC patients within a scenario of rapidly changing donor resources, such as that in the current pandemic. The model consists of the following four factors (Figure 1): (a) five-year post-LT survival (i.e., recipient outcome and utility) and mortality without LT (i.e., individual need for LT and urgency) contribute to the calculation of individual transplant survival benefit in T2 HCC patients and non-HCC patients; (b) WL mortality (i.e., expression of the population/national need for LT) and donor resources (i.e., expression of healthcare resources) contribute to the calculation of the harm to others on the WL procured by each organ allocation. These four ethical factors represent the fundamental ethical tension existing between individual and population needs with regard to LT. This ethical tension can be simply represented as a vector computation (Figure 1) where individual transplant benefit for T2 HCC patients should outweigh the harm to others on the WL to define appropriate organ allocation. Therefore, we defined the “net benefit of LT” as the difference between the individual transplant benefit and the harm to others on the WL due to each organ allocation. The overall design of this study involved analyzing the average characteristics of a nationally representative cohort of patients on the WL, and then creating a Markov model to estimate the harm to other patients on the WL caused by individual organ allocation considering different possible scenarios of decrease in donor resources. Individual five-year transplant benefit of T2 HCC patients that should outweigh the harm to others on the WL was calculated using a previous simple formula derived from a recent Italian study [6]: five-year transplant benefit T2 HCC (life months) = 1.14*MELD − 0.46*log(alpha-fetoprotein) + 0.52. A similar formula was used to obtain the five-year transplant benefit of non-HCC patients [6]: five-year transplant benefit non-HCC (life months) = 0.89*MELD − 3.59.

### 2.2. Data Source and Survival Modeling

The variables used in constructing the model were obtained from the Informative Transplant System database. This is a prospective database managed from the Italian National Transplant Centre recording prospectively collected data from each Italian center at different time points (at the moment of waiting list activation, during pre-LT follow-up visits, at the moment of LT, and during post-LT visits). Using this database, we found that 8567 adult patients with end-stage chronic liver disease (status 1 and pediatric patients were excluded) entered the WL for LT in Italy from January 2012 to December 2018: 6476 (75.6%) patients underwent LT, 1345 (15.7%) dropped out from the WL (for disease progression or death), and 746 (8.7%) still wait or were removed from the WL for disease improvement. These data were used for calculating the harm to others on the WL. WL patients were divided into subgroups: HCC patients and non-HCC patients with MELD scores of ≤20, 21–30, and >30. Italian averages for WL size and characteristics, number of organs each year, mortality rates on the list, dropout rates, and time to transplantation were obtained from this population (Table 1 and Appendix A). Follow-up data were collected up until 31 December 2019. Since under the MELD system patients at the highest risk of WL death also generally have the highest transplant rate, this potential bias was corrected through the inverse probability of censoring weighting, a well-established method to overcome dependent censoring [26]. Basically, patients with a higher (lower) probability of receiving a liver transplant are assigned a higher (lower) weight to balance out the fact that relatively less (more) follow-up on such patients is actually observed. Inverse probability of censoring weights was calculated using generalized boosted models as described by McCaffrey et al. [27], and all variables described in Table 1 were included in boosted models to balance patients undergoing transplantation or not. WL probabilities of death/dropout/transplant were obtained using Kaplan–Meier survival curves and multivariate Cox models. MELD- and HCC-related probabilities and hazard ratios (Table 1) were adjusted for sex, age, etiology of liver disease, body mass index, and blood group. WL survival curves were used to calculate daily probabilities (dp) of death/dropout or transplant according to the adjusted DEALE formula [28]: dp = 1 − e (lnS(t))/t, where t is time expressed in months and S is the survival probability.

### 2.3. Markov Decision Model

A Markov model was dedicated to the calculation of the harm to others on the WL (Appendix A) due to individual organ allocation. As in previous studies [29,30,31], the model considers that when an organ is allocated to a specific candidate, the other patients on the WL all wait for one extra organ arrival cycle and are subject to the extra risk of death during this time period. The increase in harm caused by one organ allocation, therefore, depends on the number of patients on the WL, the WL mortality rates, and the number of organs available in the unit of time. This scenario is displayed in Appendix A. The harm to others on the WL (loss in life expectancy) was calculated by subtracting the intention-to-treat five-year WL life expectancy after individual organ allocation (where WL patients will wait an extra organ arrival cycle without LT) from the intention-to-treat five-year life expectancy predictions without this allocation. Interestingly, the value of this harm is closely related to the number of organs available in the unit of time. For example, if the usual number of organs available each year is 36, the additional time other patients on the WL should wait for a transplant (and at risk of death/dropout without LT) would be about 10 days. If the number of organs is halved from a pandemic event, this additional time would increase to about 20 days and would double the harm to other patients on the WL. From this perspective, we can consider this value an effective endpoint able to describe the acute impact of a pandemic event on a liver transplant WL. Since the model aims to calculate the harm to other patients on the WL due to individual organ allocation, we assumed that transplant probabilities of other patients in the WL do not change in an acute pandemic scenario, but they are exposed to an extra risk of death only because they have to wait additional time for the next organ arrival. Average transplant probabilities in Table 1 represent, in fact, median values of a nationally representative cohort of patients on the WL who were followed up for a long period. As in previous studies [29,30,31], the WL patients were divided into subgroups with MELD scores of ≤20, 21–30, and >30. Separate Markov processes were developed for each of these groups, as well as for patients with HCC. The Markov cycle length was one day, and the time horizon was five years. The Markov model converted daily death or LT probabilities into life expectancy values. A Monte Carlo simulation was then used to understand the impact of the number of organs available in the unit of time on the model results and to estimate the level of uncertainty of such results. The uncertainty of life expectancy estimations was estimated assuming uniform distributions for the number of organs available in the unit of time, individual five-year survival rate after LT and median survival without LT, WL size (number of patients), MELD score, and different proportions of WL subgroups according to disease severity. Using the Monte Carlo simulation, we obtained a list of 10,000 outcomes of harm to others on the WL (life months) based on covariate distributions. The impact of variables on the harm to others on the WL distribution of 10,000 outcomes obtained from the Monte Carlo simulation was determined using the multivariate standard least square regression method. Statistical significance was set at *p* < 0.05. Multivariate regression served to create a simple equation that has the potential to be used in different geographical scenarios with different WL characteristics to calculate the harm to others on the WL. As in previous studies [29,30,31], we finally hypothesized that a patient with T2 HCC should receive an LT if his/her transplant survival benefit calculated using the formula from our previous study [6] is greater than the cumulative harm to the rest of the WL (Appendix A), determining a net survival benefit, as described in our ethical framework (Figure 1). In our model, we did not consider the potential impact of COVID-19 infection on post-LT mortality since it seems that this infection does not worsen the survival of transplanted patients [32]. All statistical calculations were performed using SAS (version 9.2, Cary, NC, USA), R 4.0.0 GUI 1.71 (Catalina build, 7827), and TreeAge Pro v2013 (TreAge Software, Williamstown, MA, USA).

## 3. Results

### 3.1. Estimation of Harms to Patients on the Waiting List Due to Individual Organ Allocation to a T2 HCC Patient

We first modeled a scenario of individual organ allocation to a T2 HCC patient in a situation of a usual number of organs each year; then, we modeled different scenarios of individual allocation where donor resources decreased due to competing needs of COVID-19 patients at a hospital. With a WL size of 58 patients and 49 organs arriving per year, one organ allocation increased the risk of a death occurring among the WL cohort by 42.7%. This higher mortality risk translated into a loss of 17.9 life months (Figure 2, starting point in the y axis of the curve). Using sensitivity analysis, we depicted the variation of the harm to others on the WL due to individual organ allocation in different scenarios of a decrease in the number of available donors (Figure 2). In a pandemic situation with a 50% decrease in donor resources, because of lower subsequent chances for patients to get transplanted, the impact of one organ allocation was larger. In this situation, one organ allocation transplant increased the risk of a death occurring among the WL cohort by 85.4%. This higher mortality risk translated into a loss of 35.8 life months (Figure 2).

While this aggregate harm was substantial, the harm to individual patients on the WL was much smaller and varied by MELD score and the presence of HCC, as shown in Appendix A. In order to obtain a complete assessment of the relative contribution of each covariate introduced in the model on the calculated harm to others on the WL, a Monte Carlo simulation was performed, obtaining a distribution of 10,000 outcomes, and then a multivariate regression method was used. The relative contribution of each covariate to the WL harm was measured by the T ratio value (Figure 3, Table 2). The two covariates with the highest positive impact on the harm to the WL were the percentage of organ decrease (T ratio = 94.75) and the percentage of WL patients with MELD > 20 (T ratio = 34.25). In contrast, the covariates with the highest negative impact on the harm to the WL were the ratio between the number of donors and the number of WL patients (T ratio = −18.40), and the percentage of WL patients with HCC (T ratio = −16.93). Therefore, we used only these four covariates to obtain a simplified equation with the potential to calculate the harm to others on the WL. Harm to others on the WL = Exp (3.07 + 0.03*%organ decrease + 1.18*%WL with MELD > 20 − 0.88*%WL with HCC − 1.03*N° donors/N° WL patients).

### 3.2. MELD Threshold Values and Net Transplant Benefit Estimations

Our model was finally used to determine which types of T2 HCC patients should be considered for transplantation in settings of pandemic-induced shortage, by comparing the benefit to the individual T2 HCC patient against the harm caused to other patients when that organ is used. First, using the simplified formula to calculate 5-year T2 HCC transplant benefit [6] and the simple equation obtained by the above Monte Carlo simulation, we modeled the threshold value for the MELD score to outweigh the harm to others on the WL. These values showed significant variability depending on the severity of organ shortage (Figure 4) in both T2 HCC and non-HCC patients. In Figure 4, we described the MELD score threshold values that individual candidates should outweigh in a situation of usual organ availability and in one of acute organ decrease. For example, the harm to others on the WL cohort caused by individual organ allocation was offset by the survival benefit of LT for a T2 HCC patient whose MELD score exceeded 15 (Figure 4a), in a situation of usual organ availability. This threshold increased to 24 for a non-HCC patient in a similar situation of usual organ availability (Figure 4b). In a situation of >65% acute decrease of organs in a T2 HCC, and 50% decrease in a non-HCC patient, the cumulative harm outweighed the benefit of transplantation whatever the MELD score (Figure 4). The impact of alpha-fetoprotein levels on individual 5-year transplant benefit of T2 HCC patients and consequently, on MELD threshold variation was not clinically relevant. For example, Figure 4a was constructed using an alpha-fetoprotein value of 10 ng/mL; if we used an alpha-fetoprotein value of 1000 ng/mL, the MELD threshold values decreased by less than 1 unit (i.e., the MELD threshold value became 14 with usual donor resources).

Second, we analyzed the relationship between harm to others on the WL and individual LT survival benefit in a T2 HCC patient from another point of view, that is, the estimation of life months gained (i.e., net benefit = individual benefit minus harm to others on the WL) depending on MELD score variations in different pandemic scenarios (Figure 5). As expected, the net individual transplant benefit values were much lower than those of conventional transplant benefit in a situation of usual organ availability (Figure 5). The net benefit of LT in a T2 HCC patient in a usual situation, in fact, varied from 0 life months with MELD 15 to 34 with MELD 40.

Net benefit of LT in a T2 HCC patient significantly decreased when a 50% decrease in organ availability occurred. In this situation, potentially occurring during a pandemic, the life months gained varied from 0 with MELD 30, to 12 with MELD 40.

## 4. Discussion

### 4.1. Novel Ethical Framework 

Pandemics represent dramatic challenges where competing needs have to be faced by means of ethically and functionally acceptable resource redistribution, with the aim to maintain equality of opportunities for every single individual [13]. In such settings, since scarcity is the mother of allocation, it is particularly relevant to regulate allocation according to the amount of resource reduction and time-related resource variations. In this study, we proposed a novel ethical framework to link variations in healthcare resources to individual allocation in LT using a sort of vector computation (Figure 1). By reasoning only on a population/national basis, it is possible to describe ethical tensions occurring in LT scenarios as they all contribute in the same direction [25]. In a study by Chew CA et al. [25], WL mortality, recipient outcomes, health system resources, and donor/graft safety all contributed equally in the same direction to the final quadripartite equipoise score, obtained by calculating a pyramid volume (i.e., the lower the WL mortality, and the lower the healthcare resources and recipient outcome, the lower the final score). The quadripartite equipoise score, however, is useful for the regulation of transplant activity in a general population, but it does not aid in individual clinical decision-making in LT. In contrast, our model represents the fundamental ethical tension in LT existing between individual and population needs, tensions that are directed by definition in the opposite direction. We decided to describe this ethical tension as a vector computation (Figure 1) where individual transplant benefit should outweigh the harm to others on the WL to define appropriate organ allocation. Other differences between these two ethical frameworks are that while in the quadripartite equipoise score paper [25] WL mortality (expressing the population need for LT) and healthcare resources contributed in the same direction to increase the score, in the harm to others on the WL they have an opposite direction (the lower the resources, the higher the WL mortality, the higher the harm). Moreover, recipient outcome in the Chew CA paper [25] is only expressed as five-year post-LT survival, while in our model we expressed this fundamental variable as individual transplant benefit (post-LT life expectancy minus life expectancy without LT); a very interesting principle with the potential to offer a fair distribution of resources between HCC and non-HCC patients [4,5,6]. 

### 4.2. Novel Quantitative Model Measuring the Net Benefit of LT

One important finding of this study is that individual organ allocation increases the chance of patient death by 42.7% based on the cumulative impact on the WL, and this impact is even greater if the subsequent number of available organs remains reduced (Figure 4). These data can be used when discussing with hospital administrators the pros/cons of continued transplant activity during the pandemic. In other words, although it is tempting to view a few months delay in transplantation as minimally harmful to the individual patient, the cumulative harm to the WL population is substantial. The second very important finding is that, given at least some unavoidable decrease in transplant activity, the remaining transplant opportunities need to be focused on patients with the highest net survival benefit. The model shows how, particularly in a time frame scenario where the pandemic negative impact on transplant activities would persist for months (i.e., in presence of infection outbreak rebounds due to variants), the risk of selecting patients with a net harm to others on the waiting list rather than a net benefit is more than concrete. In such a scenario, T2 HCC patients with compensated cirrhosis (i.e., low MELD score) have a relatively good middle-term survival without transplantation due to the availability of valid therapeutic alternatives to LT [33,34]. For these patients, the expected individual transplant benefit is not able to warrant a net benefit over the harm to others on the waiting list. Therefore, these patients should be excluded from transplantation in a phase of significant donation crisis, such as the one we had during the second phase of the pandemic, due to virus variants. On the contrary, T2 HCC patients with decompensated cirrhosis (i.e., mid–high MELD values) have very low survival perspectives without LT, thus, their net transplant benefit guarantees a good indication to LT also in a pandemic scenario. This analysis corroborates common-sense practice empirically adopted in many centers: the greater the proportion of transplant activity reduction (and the longer the protraction of donor scarcity), the more the selection and allocation directed to patients without therapeutic alternatives and with better post-transplant life expectations [14]. More generally, a quantitative approach to evaluating the trade-off between the need for a transplant, expected post-transplant survival, and harm to the WL is crucial for converting subjective speculations into a more objective analysis, particularly in conditions such as the present pandemic, where resource scarcity occurs suddenly and is subjected to potentially rapid time- and regional-dependent variations. Our model does not want to contraindicate individual LT in situations of severe organ shortage. In fact, it is obvious that using organs is always better than not using them, even in an imperfect manner. Our model simply suggests that the main allocation focus should be on candidates with the highest transplant benefit, especially in situations of organ shortage. Additionally, when the maximum organ decrease is reached (65% for a T2 HCC and 50% for a non-HCC patient, as in Figure 4), our model results do not mean that an LT program should stop, but simply indicate that a specific LT program has reached a dramatic situation where its waiting list characteristics, in terms of the number of patients waiting for LT and severity of their liver disease, are not sustainable anymore with such a scarce number of available organs. In other words, when organ decrease is so relevant to determine a “net harm” for each single organ allocation, there is not the possibility to further optimize utility and transplant benefit principles. In such dramatic situations, conversely, LT centers should privilege only the urgency principle dedicating scarce organ resources only to patients on immediate clinical need who would otherwise die without a transplant in a short period of time (e.g., acute liver failure, very high MELD, etc.). 

### 4.3. Other Potential Applications of Our Model

Previously published studies used a similar model [29,30] to face the issue of geographical differences in donor availability. The model here presented, on the contrary, suggests an intraregional or even an intra-center modulation of allocation policies, prompted by sudden context changes. Whether this intra-center modulation is ethically acceptable may be a matter of debate as well, but the exceptional and unprecedented characteristics of the COVID-19 pandemic may, at least partially, justify such a time/context-related policy change. Furthermore, the model provides cut-offs of transplant activity reduction at which the harm on the WL is greater than the benefit obtained by the transplant itself, making the whole procedure counter-productive. These limits, particularly in a prolonged crisis scenario, may serve as benchmarks to guide intra-hospital resource distribution (namely, procurement-dedicated personnel, and ICU beds) between viral outbreak and transplant-related activities. Finally, using the COVID-19 scenario, the present model has the potential to provide information even in an opposite condition where donor resources durably increase in a certain geographic/organizational setting, allowing modulations of selection/allocation criteria based on a quantitative framework analysis [2]. 

One of the main advantages of our model is that its use is not limited to an era of global pandemic, but could aid allocation policies also in situations of normal or increased donor resources. MELD score thresholds in both HCC and non-HCC patients (Figure 4 and Figure 5) could be used to aid eligibility and allocation phases to exclude patients with a negative net benefit from being listed or allocated organs.

### 4.4. Potential Limitations of the Presented Model

The present study has several potential limitations. First, the model is based on the recent Italian liver transplant population data. Even though epidemiologically similar to many south European countries, the referral population may reveal important differences with other transplant settings either in terms of patient characteristics or in regard to donor availability, waiting time, and list management. The use of Monte Carlo simulations including confidence interval variations, however, has the potential to partially overcome this limit, also favoring a useful external application of our prediction models. Second, the choice of five-year post-transplantation as a horizon for evaluating survival benefit was arbitrary, and a 10-year horizon scenario would probably have resulted in a more inclusive policy. However, in our opinion, critical situations, such as the one produced by the COVID-19 pandemic, justify the choice of a shorter outlook, more in line with the need for balanced resource redistribution in an emergency phase. Third, we acknowledge that translating these results into clinical behaviors may be difficult in the context of complex multiregional, national, or multi-national organizations (i.e., UNOS and Eurotransplant) in which policy changes need to go through articulated processes and are finalized after public appraisal. However, we believe the availability of quantification models, even though necessarily inaccurate with respect to every particular population, may promote urgent policy adjustments at central levels and, more importantly, may also provide clinicians with a more solid guide in patient selection at listing. Finally, our mathematical model, originally designed for LT, could not be used for other SOTs. On a purely speculative basis, however, the main theoretical findings of our study (i.e., supporting the continuation of transplantation during the COVID-19 crisis with a particular focus on those patients with the highest survival benefit) could also be extended to other SOTs.

## 5. Conclusions

In conclusion, our study supports the continuation of the transplantation of T2 HCC patients during crises such as COVID-19; however, the focus needs to be on those patients with the highest net survival benefit. In particular, only T2 HCC patients with decompensated cirrhosis and without therapeutic alternatives should be considered for LT during a phase of acute organ shortage. To provide quantitative measures at different degrees of transplant activity decrease, we propose a flexible and quantitative model to couple the amount of harm to others on the WL to the expected benefit in different time frames and donor-scarcity scenarios. Analytic frameworks such as the one proposed here may be of help to transplant clinicians in patient selection processes throughout phases of rapid change in donor resource and transplant activity.

## Figures and Tables

**Figure 1 cancers-13-01416-f001:**
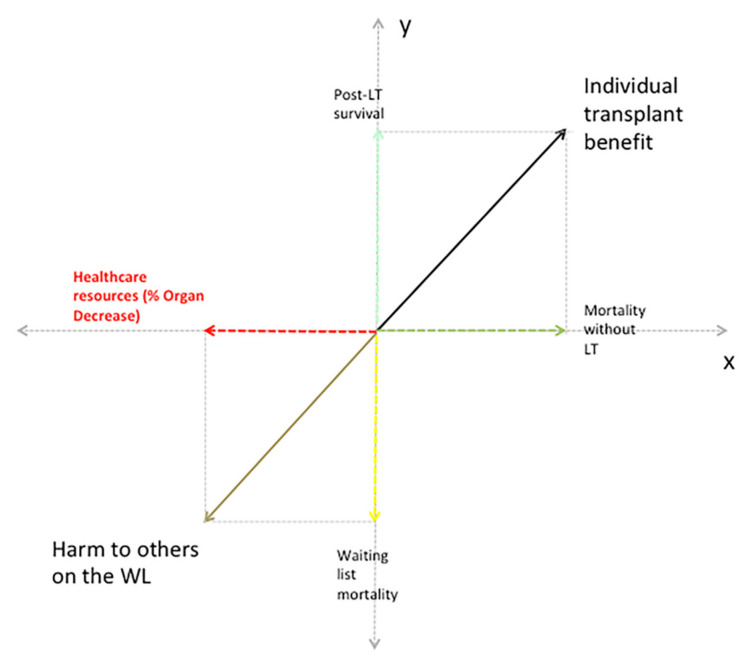
Ethical framework for describing the ethical tension between healthcare resources and individual transplant benefit in hepatocellular carcinoma (HCC) patients. The equipoise between individual and population needs is determined by the vector computation of individual transplant benefit (black) minus harm to others on the waiting list (brown). The individual transplant benefit is determined by the vector computation of individual post-LT survival (light blue) and mortality without transplant (green), while the harm to the others on the waiting list is determined by the vector computation of healthcare resources (i.e., % organs decrease, red) and population waiting list mortality (yellow). Abbreviations: LT, liver transplantation; WL, waiting list.

**Figure 2 cancers-13-01416-f002:**
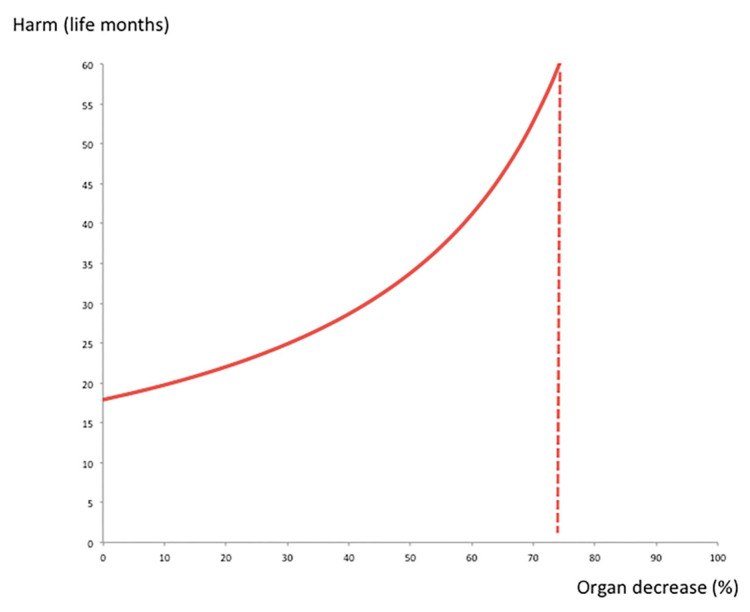
Sensitivity analysis applied to the Markov model. Impact of a decrease (%) in organ availability on the aggregate harm (months) to others on the waiting list due to individual organ allocation in T2 HCC patients.

**Figure 3 cancers-13-01416-f003:**
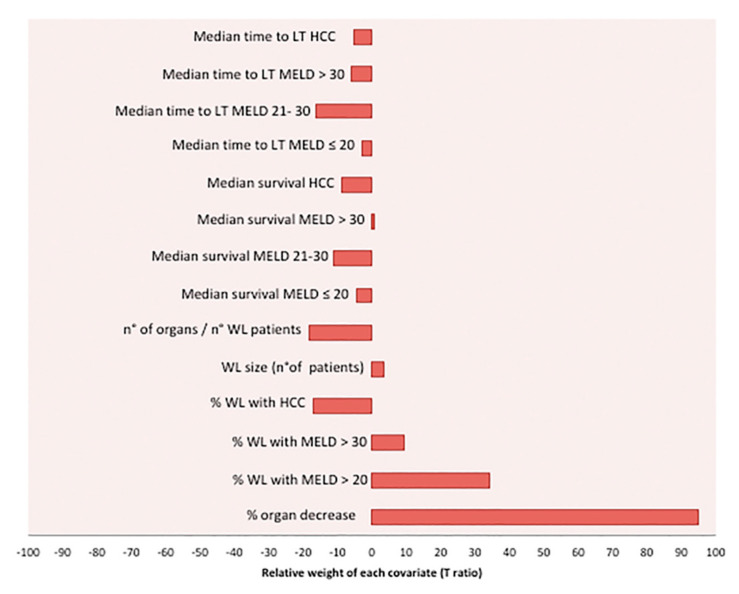
Multivariate analysis on the 10,000 outcome results of the Monte Carlo simulation. The relative contribution of each covariate to the harm to others on the WL was measured by the T ratio value of each covariate distribution introduced in the Monte Carlo simulation.

**Figure 4 cancers-13-01416-f004:**
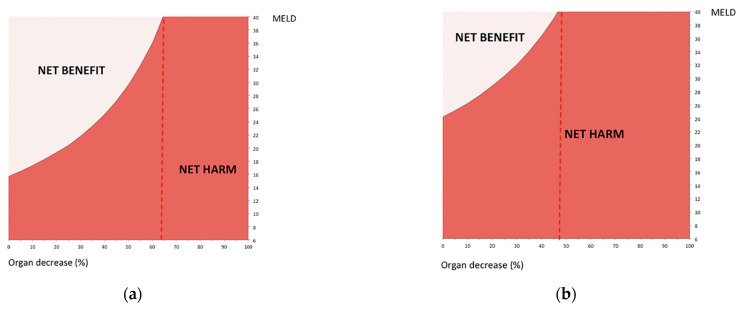
Impact of acute organ shortage due to COVID-19 pandemic on MELD score threshold values to decide organ allocation in T2 HCC (**a**) and non-HCC (**b**) patients.

**Figure 5 cancers-13-01416-f005:**
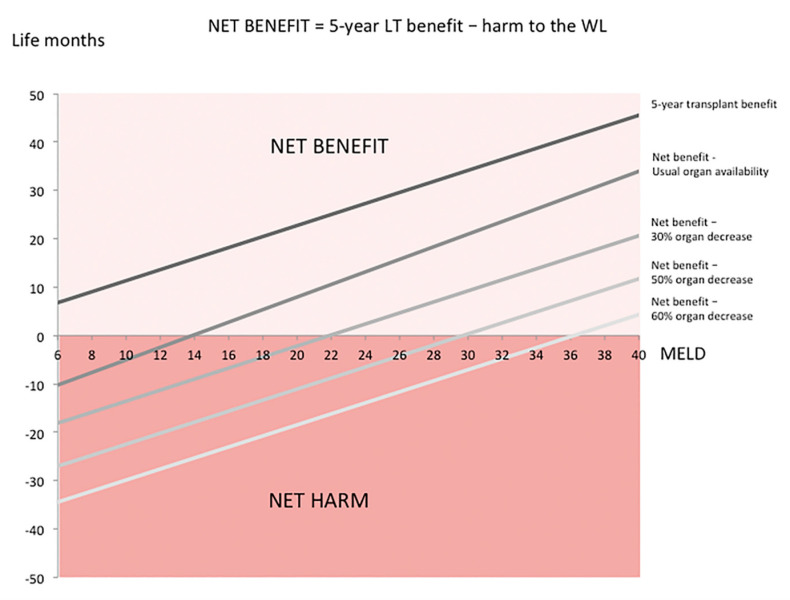
Net benefit (individual transplant benefit minus harm to others on the WL) estimation of LT in a T2 HCC patient in situations of usual organ availability and 50% decrease in organ availability compared to individual five-year transplant benefit estimation.

**Table 1 cancers-13-01416-t001:** Variables utilized in constructing the Markov model.

Variables	Italian Average Center	Range Tested
WL size (number)	58	46–70
Organs arriving per year (number)	49	39–59
Median survival in the WL (months)		
MELD ≤ 20	29.5	27.9–32.0
MELD 21–30	6.7	6.0–8.1
MELD > 30	0.9	0.8–1.6
HCC	20.7	18.6–21.7
Median time to transplant (months)		
MELD 11–20	12.8	12.1–14.0
MELD 21–30	4.4	3.9–5.1
MELD > 30	0.7	0.5–1.3
HCC	9.5	8.9–10.2
WL stratification according to disease severity (%)		
MELD ≤ 20	37	30–45
MELD 21–30	13	10–15
MELD > 30	4	3–5
HCC	46	37–55
5-year post-LT survival of WL patients (%)		
MELD ≤ 20	83	75–90
MELD 21–30	81	73–90
MELD > 30	76	68–84
HCC	81	72–90

Abbreviations: WL, waiting list; MELD, model for end-stage liver disease; HCC, hepatocellular carcinoma; LT, liver transplantation.

**Table 2 cancers-13-01416-t002:** Multivariate analysis showing the contribution of each covariate introduced in the Monte Carlo simulation on the harm to others on the WL. Positive values increase the harm to the WL, while negative values decrease it. Using a simple equation including some of the estimates, it is possible to calculate the harm to others on the WL.

Variables	Estimate	Standard Error	T Ratio	*p*-Value
Constant	3.7228	0.0149	8.99	<0.0001
% Organs decrease	0.0291	0.0004	94.75	<0.0001
N° of organs/N° of WL patients	−1.0355	0.0078	−18.40	<0.0001
WL size (N° of patients)	0.0045	0.0002	3.46	0.0005
% WL with MELD > 20	1.1601	0.0049	34.25	<0.0001
% WL with MELD > 30	1.8280	0.0277	9.29	<0.0001
% WL with HCC	−0.8624	0.0078	−16.93	<0.0001
Median survival MELD ≤ 20	−0.0041	0.0001	−4.43	<0.0001
Median survival MELD 21–30	−0.0336	0.0004	−11.06	<0.0001
Median survival MELD > 30	−0.0014	0.0030	0.81	0.4179
Median survival HCC	−0.0075	0.0001	−8.73	<0.0001
Median time to LT MELD ≤ 20	−0.0055	0.0002	−2.85	0.0044
Median time to LT MELD 21–30	−0.0484	0.0004	−16.31	<0.0001
Median time to LT MELD > 30	−0.1181	0.0028	−6.03	<0.0001
Median time to LT HCC	−0.0066	0.0002	−5.27	<0.0001

Abbreviations: WL, waiting list; MELD, model for end-stage liver disease; HCC, hepatocellular carcinoma. Simplified equation to calculate the harm to others on the WL obtained using the estimates of the strongest covariates: harm to others on the WL = Exp (3.07 + 0.03*%organ decrease + 1.18*%WL with MELD > 20 − 0.88*%WL with HCC − 1.03*N° donors/N° WL patients).

## Data Availability

Restrictions apply to the availability of these data. Data were obtained from the Informative Transplant System database. This is a prospective database managed by the Italian National Transplant Centre (directly from the Italian Health Ministry) where prospectively collected data from each Italian center are recorded at different time points. The management of this database conforms to the Italian legislation on privacy. Data are available only with the permission of the Italian National Transplant Centre.

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
