# Peer review of "Liver Transplantation for T2 Hepatocellular Carcinoma during the COVID-19 Pandemic: A Novel Model Balancing Individual Benefit against Healthcare Resources"

_cancers, 2021, doi:10.3390/cancers13061416_

Round 1
Reviewer 1 Report
During Covid 19 pandemic period, the net benefit of the liver transplantation in the patients with T2 HCC was calculated by Markow model using an Italian national prospective database. The authors concluded that liver transplantation for the patients with T2 HCC has the highest benefit also in the present Covid 19 pandemic era.
I have some comments.
- (Simple summary) The abbreviations should be appeared with the spelled words when appeared for the first time in summary, HCC and MELD.
- (Abstract) The abbreviations should be appeared with the spelled words when appeared for the first time in summary, HCC and MELD.
- (Introduction) It consists of a unstructual many sentences, which is difficult for the reviewer to follow. It should be divided into some paragraphs. I recommend the second starting from “(L75) The need to link “, the third from “ (L92) Common sense …” and forth from “(L106) The aim of this study …”.
- (Discussion) It consists of a unstructual many sentences, which is difficult for the reviewer to follow. I recommend the second starting from “(L348) One important findings … “, the third from “ (L381) The model here presented ,…” and forth from “(L394) One of the main …”.
- (L340) What is QE score?
Author Response
During Covid 19 pandemic period, the net benefit of the liver transplantation in the patients with T2 HCC was calculated by Markow model using an Italian national prospective database. The authors concluded that liver transplantation for the patients with T2 HCC has the highest benefit also in the present Covid 19 pandemic era.
I have some comments.
- (Simple summary) The abbreviations should be appeared with the spelled words when appeared for the first time in summary, HCC and MELD.
ANSWER: As suggested, in the revised paper we have now spelled all abbreviations when appeared for the first time in simple summary.
- (Abstract) The abbreviations should be appeared with the spelled words when appeared for the first time in summary, HCC and MELD.
ANSWER: As suggested, in the revised paper we have now spelled all abbreviations when appeared for the first time in the Abstract.
- (Introduction) It consists of a unstructual many sentences, which is difficult for the reviewer to follow. It should be divided into some paragraphs. I recommend the second starting from “(L75) The need to link “, the third from “ (L92) Common sense …” and forth from “(L106) The aim of this study …”.
ANSWER: We have now divided the Introduction in the suggested paragraphs.
- (Discussion) It consists of a unstructual many sentences, which is difficult for the reviewer to follow. I recommend the second starting from “(L348) One important findings … “, the third from “ (L381) The model here presented ,…” and forth from “(L394) One of the main …”.
ANSWER: We have now divided the Discussion in the suggested paragraphs.
- (L340) What is QE score?
ANSWER: The quadripartite equipoise (QE) score was obtained calculating a pyramid volume that considers the four ethical factors (i.e. quadripartite) involved in liver transplantation allocation: WL mortality, recipient outcome, health system resources, and donor/graft safety. This score was described in a recent paper (Chew CA, Iyer SG, Kow AWC, Madhavan K, Wong AST, Halazun KJ, et al. An international multicenter study of protocols for liver transplantation during a pandemic: A case for quadripartite equipoise. J Hepatol 2020; 73: 873-881.). We have now removed this misleading abbreviation from our revised paper using only the full length name of “quadripartite equipoise score”.

Reviewer 2 Report
Overall Comments:
This paper offers a novel, thoughtful, and reasonable analysis of a difficult problem in the COVID era. Overall, the paper appears methodologically sounds and conceptually appropriate. The application of a Markov model to this problem, though not entirely new, is appropriate and easily interpretable to a clinician. The authors should be commended for attempting to determine the benefit of a transplant to a specific subpopulation vs. the general waitlist in which the allocation strategies for these the subpopulation vs. the general population are different. This kind of problem exists in multiple areas of transplantation.
However, multiple issues must be addressed prior to publication. First, all figures need to be updated so that they are readable/interpretable. In being converted to a PDF, a number of the figures have lost information/are unreadable. Next, the paper would benefit from extensive English language editing. There are multiple points where the manuscript uses incorrect verb tenses and grammar that makes the interpretation difficulty. Finally, the authors could create more subsections in their discussion. As currently written, it is difficult to follow.
Specific Comments:
- Simple Summary: This is too long and is essentially a restatement of the abstract. I would favor shortening this to 3-4 sentences to convey the main messages of the paper.
- Methods:
- Organ “arrival” is an odd phrase and should be either defined or revised. I would favor defining very explicitly as there is not a perfect term to capture this concept widely in use.
- Results
- As above, all figures have errors in the labeling of axes/other lables. Please correct
- Figure 3
- Please do not overlap the text with the bars or make the text more easily readable within the bars
- In many places the authors utilize either multivariate or multivariable. Certainly the Markov model incorporated time into the calculation so it seems that multivariate is appropriate (time varying co-variates) but please choose one and be consistent.
- Table 2
- T-scores do not seem relevant to the standard reader. P-values are sufficient. Please remove these
- Figure 4
- Please explain the shaded vs. unshaded areas more explicitly. My understanding is that the shaded region is where there is no benefit to transplant at a given MELD and the unshaded is where there is no benefit (or even a detriment) at a given MELD, however the curves don’t exactly align with that expectation. It is also non-intuitive how there can be such a severe shortage that transplanting anyone leads to a decrease in survival for everyone on the WL. Certainly not utilizing the organs would be worse than utilizing them, even in an imperfect manner. Please explain this more thoroughly.
- Figure 5
- It may be nice to graph this in such a way as to show the negative impact of transplantation of T2-HCC on the WL at higher scarcity levels (i.e. extend graph into the cartesian quadrand 3).
- As above, please create subheadings for your discussion. This section also requires the most English language editing.
Author Response
Overall Comments:
This paper offers a novel, thoughtful, and reasonable analysis of a difficult problem in the COVID era. Overall, the paper appears methodologically sounds and conceptually appropriate. The application of a Markov model to this problem, though not entirely new, is appropriate and easily interpretable to a clinician. The authors should be commended for attempting to determine the benefit of a transplant to a specific subpopulation vs. the general waitlist in which the allocation strategies for these the subpopulation vs. the general population are different. This kind of problem exists in multiple areas of transplantation.
ANSWER: We really thank the reviewer for these positive comments.
However, multiple issues must be addressed prior to publication. First, all figures need to be updated so that they are readable/interpretable. In being converted to a PDF, a number of the figures have lost information/are unreadable. Next, the paper would benefit from extensive English language editing. There are multiple points where the manuscript uses incorrect verb tenses and grammar that makes the interpretation difficulty. Finally, the authors could create more subsections in their discussion. As currently written, it is difficult to follow.
ANSWER: We sincerely thank the reviewer for these suggestions giving us the possibility to improve our manuscript. In the revised paper we have updated our figures according to reviewer’s suggestions and uploaded high quality figures in order to increase their readability and interpretability. Moreover we have performed a complete English language revision using the MDPI English editing service where “all manuscripts are edited by native speakers”. Finally we have created new subsections in the discussion.
Specific Comments:
- Simple Summary: This is too long and is essentially a restatement of the abstract. I would favor shortening this to 3-4 sentences to convey the main messages of the paper.
ANSWER: As suggested, we have now shortened the simple summary giving only the main messages of the paper.
- Methods:
- Organ “arrival” is an odd phrase and should be either defined or revised. I would favor defining very explicitly as there is not a perfect term to capture this concept widely in use.
ANSWER: As suggested, we have now very limited the used of “organ arrival” in the revised paper. In the great majority of cases we now preferred to define more explicitly the same concept using for example “number of organs available in the unit of time or each year”. In some circumstances, other definitions such as “donor resources” or “organ availability” were used to better explain the same concept of “organ arrival”.
- Results
- As above, all figures have errors in the labeling of axes/other lables. Please correct
ANSWER: As suggested, we have corrected all errors in the figures.
- Figure 3
- Please do not overlap the text with the bars or make the text more easily readable within the bars
ANSWER: As suggested, we have corrected all errors in figure 3.
- In many places the authors utilize either multivariate or multivariable. Certainly the Markov model incorporated time into the calculation so it seems that multivariate is appropriate (time varying co-variates) but please choose one and be consistent.
ANSWER: As suggested, in the revised paper we have now used only the term multivariate.
- Table 2
- T-scores do not seem relevant to the standard reader. P-values are sufficient. Please remove
ANSWER: We understand the reviewer’s perplexity about T-ratio values in table 2. However, these values give a better idea of the relative contribution of different covariates to the harm to others on the WL. In fact, in Figure T ratio values are reported. We kindly ask the reviewer to accept we maintain this parameter in Table 2 to maintain a inter-manuscript coherence between table 2 and figure 3.
- Figure 4
- Please explain the shaded vs. unshaded areas more explicitly. My understanding is that the shaded region is where there is no benefit to transplant at a given MELD and the unshaded is where there is no benefit (or even a detriment) at a given MELD, however the curves don’t exactly align with that expectation. It is also non-intuitive how there can be such a severe shortage that transplanting anyone leads to a decrease in survival for everyone on the WL. Certainly not utilizing the organs would be worse than utilizing them, even in an imperfect manner. Please explain this more thoroughly.
ANSWER: We thank the reviewer for this comment giving us the possibility to improve our Figure 4. We have now labeled the shaded and not-shaded areas of the figure using “Net benefit” and “Net harm”. Moreover we have added a red dashed line to indicate the maximum organ decrease above which the cumulative harm to the WL due to a single organ allocation during a pandemic outweighed the benefit of transplantation whenever MELD score. We understand reviewer’s perplexity about these thresholds values. We agree with the reviewer on the fact that “using organs is always better than not using them even in an imperfect manner”. However, our model does not want to contraindicate individual LT in situations of severe organ shortage. In fact, it is obvious that using organs is always better than not using them, even in an imperfect manner. Our model simply suggests that the main allocation focus should be on candidates with the highest transplant benefit, especially in situations of organ shortage. Additionally, when the maximum organ decrease is reached (65% for a T2 HCC and 50% for a non-HCC patient, as in Figure 4), our model results do not mean that an LT program should stop, but simply indicate that a specific LT program has reached a dramatic situation where its waiting list characteristics, in terms of number of patients waiting for LT and severity of their liver disease, are not sustainable anymore with such a scarce number of available organs. This fact could suggest to LT centers that in such a dramatic situation, patients should not be added to the WL and scarce organ resources should be directed only to more urgent patients already in the WL. We added some of these concepts in the revised discussion.
- Figure 5
- It may be nice to graph this in such a way as to show the negative impact of transplantation of T2-HCC on the WL at higher scarcity levels (i.e. extend graph into the cartesian quadrand 3).
ANSWER: Again we thank the reviewer for this comment giving us the possibility to improve our Figure 5. We have now represented also “net harm” values extending our graph into the fourth Cartesian quadrant
- As above, please create subheadings for your discussion. This section also requires the most English language editing.
ANSWER: We have now created subheadings for the discussion, as suggested. Moreover we have performed a complete English language revision using the MDPI English editing service where “all manuscripts are edited by native speakers”.

Round 2
Reviewer 2 Report
The authors have extensively revised their paper based on my previous suggestions and should be commended for their thorough review, including use of the English Language Editing services.
The only continued revision I would suggest is with regards to the interpretation of the "maximum organ decrease above which the cumulative harm to the WL due to a single organ allocation during a pandemic outweighed the benefit of transplantation whenever MELD score." I think that the correct interpretation here is perhaps that the allocation of organs should only be based on immediate clinical need. That is, there is no way to optimize utility, only stave off death for those that would otherwise die without transplant in a short period of time (e.g. acute liver failure, very high MELD, etc.).
Author Response
The authors have extensively revised their paper based on my previous suggestions and should be commended for their thorough review, including use of the English Language Editing services.
Answer: We thank the reviewer for these positive comments.
The only continued revision I would suggest is with regards to the interpretation of the "maximum organ decrease above which the cumulative harm to the WL due to a single organ allocation during a pandemic outweighed the benefit of transplantation whenever MELD score." I think that the correct interpretation here is perhaps that the allocation of organs should only be based on immediate clinical need. That is, there is no way to optimize utility, only stave off death for those that would otherwise die without transplant in a short period of time (e.g. acute liver failure, very high MELD, etc.).
Answer: We thank the reviewer for this criticism giving us the possibility to explain better this critical aspect. In the revised paper we add this sentence in the discussion that we think reflects the reviewer’s correct interpretation: “ In other words, when organ decrease is so relevant to determine a “net harm” for each single organ allocation, there is not the possibility to further optimize utility and transplant benefit principles. In such dramatic situations, conversely, LT centers should privilege only the urgency principle dedicating scarce organ resources only to patients on immediate clinical need who would otherwise die without transplant in a short period of time (e.g. acute liver failure, very high MELD, etc.)”.
